# CognoStroke: Automated Cognitive and Mood Assessment on the Hyper-Acute Stroke Unit

**DOI:** 10.3390/healthcare13222885

**Published:** 2025-11-13

**Authors:** Simon M. Bell, Bahman Mirheidari, Kirsty A. C. Harkness, Emma Richards, Mary Sikaonga, Madalina Roman, Jonathan Gardner, India Lunn, Isabela Ramnarine, Udit Gupta, Hamish Patel, Larissa Chapman, Katie Raine, Caitlin Illingworth, Dorota Braun, Heidi Christensen, Daniel J. Blackburn

**Affiliations:** 1Sheffield Institute for Translational Neuroscience, School of Medicine and Population Health, University of Sheffield, 385a Glossop Rd, Broomhall, Sheffield S10 2HQ, UK; 2NIHR Sheffield Biomedical Research Centre, University of Sheffield, Sheffield S10 2JF, UK; 3Neuroscience Institute, University of Sheffield, Firth Court, Sheffield S10 2TN, UK; 4Department of Neurology, Sheffield Teaching Hospitals NHS Foundation Trust, Sheffield S10 2JF, UK; 5School of Computer Science, University of Sheffield, Portobello, Sheffield S10 4DP, UK

**Keywords:** stroke, cognostroke, cognitive impairment, mood, depression, large language model

## Abstract

**Highlights:**

**What are the main findings?**
Automated cognitive and mood assessment is feasible in the acute stroke setting;Computer literacy, anxiety in using technology, and post-stroke fatigue are barriers to performing automated cognitive assessment in the acute stroke setting.

**What are the implications of the main findings?**
The identification of mood and cognitive difficulties is possible at the earliest stages post stroke;Short assessment and improving access to computers on the stroke pathway would aid further automated assessment.

**Abstract:**

**Background:** Cognitive and mood impairments are common in Stroke Survivors (SSs), leading to worse outcomes and poorer quality of life measures. Current methods of assessment of mood and cognitive performance are time consuming and rely on health care professionals. This makes assessment in hyper-acute stroke units (HASU) difficult. Here we describe the use of CognoStroke, an automated assessment of mood and cognitive impairment in the HASU. **Methods**: Using conversational interaction delivered through a virtual, web-based agent (CognosStroke), speech analysis was performed using three large language models (GPT2, Facebook.BART-based, and RobERTa-base) to classify thresholds levels of MoCA (threshold: 22,23,24,25,26), GAD-7 (above 5 and 10), and PHQ-9 (above 5 and 10). Results are presented as Macro F1-scores (MFSs). Patients were asked about barriers to using CogonStroke. **Results**: A total of 151 SSs agreed to perform CognoStroke, with 75 completing the full assessment. The best MFS of 0.723 was achieved using CognoStroke for thresholding a MoCA of 26. The MFS improved further to 0.783 when single prompts or a smaller combination of prompts from the CognoStroke bank were used. For the PHQ-9 a MFS of 0.686 was achieved thresholding above 10 and on the GAD-7 a MFS of 0.617 was achieved for thresholding above 5. Single prompts or smaller prompt combinations again achieved higher MFSs. **Discussion**: CognoStroke has potential to classify SSs into groups with high or low cognitive and mood thresholds, highlighting benefits for improving post-stroke cognitive assessment. Challenges of automated assessment on the HASU include patient computer access, anxiety in using technology, post-stroke fatigue, and computer literacy.

## 1. Introduction

There are approximately 100,000 strokes a year in the UK and 1.3 million current Stroke Survivors (SSs) [1,2]. Post-stroke cognitive and mood disorders are common and have significant effects on quality of life [3,4]. SSs with cognitive impairment have worse outcomes, are more impaired on measures of the activities of daily living, and are often unable to return to prior social and occupational roles [5,6]. At least 20% of all people that have a stroke develop long-term memory problems [7].

Low mood and anxiety occur in one in every three SSs and can make memory symptoms worse [8,9]. This is important as it compounds the disability in SSs [10]. Mood and memory problems are even found in people who have the most minor form of stroke-like symptoms and can often remain hidden on initial assessment [11]. Treatments such as cognitive rehabilitation are more effective if cognitive and mood disorders are detected early [12,13].

Stroke units are busy and challenging places to assess cognition and mood as it requires specialised practitioners who perform multiple assessments covering motor, speech, swallowing function, secondary prevention, as well as mood and cognition testing. In the UK, the Sentinel Stroke National Audit Programme highlights that even though cognition and mood assessments are part of the national audit, 8% of people do not receive a mood or cognitive assessment [2]. This means approximately 8000 SSs each year are not assessed, and 1600 SSs with cognitive impairment are potentially missed. This figure varies between stroke units, with some centres performing assessments in only 40% of SSs [2]. Missed assessments will likely increase due to early supported discharge and reduced face-to-face time with clinicians [2]. 

Automating the process of both mood and cognitive assessment has the potential to increase screening rates in SSs and provide cognitive input in resource-low services.

We have developed CognoStroke, a fully automated assessment of cognition and mood that identifies changes in speech patterns (https://www.cognospeak.com/cognostroke (accessed on 6 November 2025)). CognoStroke comprises a set of 15 memory-probing questions assessing cognitive and mood function. Preliminary data using 55 SSs who undertook CognoStroke on a hyper-acute stroke unit (HASU) has shown a good accuracy of CognoStroke classifying Montreal Cognitive Assessment (MoCA) scores into groups above and below a threshold of 26 [14]. In this paper we have expanded this initial preliminary dataset from 55 to 68 SSs. In addition, we have performed a completely new analysis using three large language models (LLMs) (GPT2 (decoder-only), Facebook.BART-based (encoder–decoder), and RobERTa-base (encoder-only)). This has led to a more complex speech pattern analysis and an ability to classify MoCA scores at several thresholds. We have also performed a completely novel analysis of CognoStroke’s ability classify mood scores using the Generalised Anxiety Disorder Assessment (GAD-7) and Patient Health Questionnaire (PHQ9) at thresholds above 5 and above 10. Furthermore, we describe the challenges that need to be considered for future versions of automated cognitive and mood assessment in acute stroke settings.

CognoStroke for the first time demonstrates the feasibility of an end-to-end approach to classifying cognitive and mood scores from conversational speech by directly fine-tuning a state-of-the-art LLM for the challenging subject of acute stroke patients. This breaks away from the conventional multi-stage pipeline (ASR → feature extraction → classifier) [14] and explores the possibility of more integrated and powerful automated neuropsychological assessment, providing a new methodological foundation for objective and scalable remote mental health assessment.

Furthermore, by comparing LLMs with different architectures—decoder-only (GPT2), encoder–decoder (BART), and encoder-only (RoBERTa)—this study provides valuable empirical insights into which architectures are effective in clinical score classification tasks based on conversational speech.

## 2. Materials and Methods

### 2.1. Ethical Approval

Ethical approval was granted by the Camberwell St Giles Research Ethics Committee in London. Approval for the study was granted on the 26th of May 2020 (REC reference: 20/LO/0376). Consent to participate in the study was obtained either electronically or on paper before the participant proceeded to the actual assessment. The completed e-consent was saved as a PDF. The e-consent and all data generated as part of the CognoStroke study was stored on a University of Sheffield Encrypted Google Drive and access only given via a password protected university account. This is in accordance with the University of Sheffield’s Acceptable Use of Data policy.

Data was collected and retained in accordance with the General Data Protection Regulation (GDPR). A database of recruited participants collated neuropsychology, mood and anxiety scores and final clinical diagnosis. The database also includes medication and co-morbid diagnoses as these can often affect the speech of participants. No patient-identifying information was stored on the database or within data files produced or in file names; only the patient alphanumeric codes were used to distinguish participants.

All participants consented for Artificial Intelligence (AI) data use and data was collected in accordance with UK Data Protection Act requirements.

### 2.2. Study Design

This study was a feasibility pilot study.

### 2.3. Study Participants

A SS was diagnosed if a stroke or Transient Ischemic Attack (TIA) diagnosis was confirmed by a clinical review and brain imaging. Those with a prior history of dementia, severe dysphasia, and limited English were not eligible to participate in this study.

SS patients that met eligibility criteria were provided with an information sheet and talked through the study. A small number of stroke patients were recruited after they had been discharged from hospital, via letter. Consent to contact forms were provided and those who consented were contacted via email with an information sheet. A time was arranged for those that provided consent, where they could ask further questions and confirm that they were willing to continue with the study. Patients were asked to complete the CognoStroke assessment alone.

### 2.4. Data Collection Procedure

Cognitive performance, mood, and anxiety levels were assessed using GAD-7, PHQ-9, and MoCA or telephone (t-MoCA) scales. Each of these assessments is a standard pen and paper-based assessment of anxiety, mood, and cognition performed in cognitive and stroke-type clinics. The GAD-7 and PHQ-9 consist of 7 and 9 questions, respectively, that the participant answers using a numerical scale from 0 to 3. The MoCA and t-MoCA assess different aspects of cognitive performance. The cut-off for what is considered an abnormal score for cognitive dysfunction after stroke is debated, with scores between 20 and 27 for the MoCA quoted [15,16]. t-MoCA scores were scaled to fit full MoCA results, prior to analysis, using a standard conversion as detailed previously (2.49 + 0.028 × t-MoCA + 0.011 × education in years) [17]. After performing these assessments, a numerical value for each of anxiety, mood, and cognitive level is available. For both the GAD-7 and PHQ-9, an overall score of above 10 is considered a moderate level of anxiety or depression. For this paper we decided to determine if the CognoStroke system could classify GAD-7 and PHQ-9 levels above 5 and above 10 as these are considered mild and moderate scores, respectively. We decided to use several MoCA score classifiers (22–26) due to the debate in the stroke literature as to what is an appropriate cut-off score to identify cognitive impairment.

All participants answered 15 standardised questions using the CognoStroke programme. All participants answered the same questions in the same order. Audio and visual responses were recorded using both laptop and desktop-based systems. The environment in which both sound and video recordings were made was not controlled to reflect the natural ward-based environment in which CognoStroke is most likely to be used. We attempted to keep noise to a minimum, but to reflect the real-world environment in which the eventual CognoStroke system would be deployed, no measures were included to control background sound distortion of recordings.

### 2.5. Speech Processing

In our previous automated pipeline, Mirheidari et al. 2024 [14], we used a Wav2vec2 automatic speech recognition (ASR) system (with a 23% word error rate (WER)) to transcribe the audio recordings. These transcriptions were then passed to a BERT model to generate embeddings, which represent features capturing the context and meanings of response. These embeddings were afterwards classified using conventional classifiers, such as Random Forest, Logistic Regression, and Support Vector Machines. The best classification results were achieved by fusing acoustic features of eGeMAPS and BERT-embedding features. In the current study we replaced the ASR component with a large Whisper model (with a lower WER of around 13%). The 15 questions that form the CognoStroke system were split into 16 separate prompts during speech processing.

### 2.6. Model Development

Compared to our previous work, we also replaced BERT embeddings with fine-tuning LLMs with a classifier head to adapt to our dataset. We did not experiment with fusion techniques in this study, as modern LLMs have been used alone on various classification tasks with reasonably good accuracy.

Three LLMs (GPT2 (decoder-only), facebook.BART-base (encoder–decoder), and RobERTa-base (encoder-only)) were used to classify levels above and below cut-off scores for the different cognitive and mood tests (MoCA thresholds of 22, 23, 24, 25, and 26), GAD-7 (above 5 and above 10) and PHQ-9 (above 5 and above 10).

For fine-tuning the LLMs with the classifier head (BART, GPT2, and ROBERTa with the sequence length to 1024, 1024, and 512, respectively), the hyperparameters were the learning rate of 2 × 10^−5^ with weight decay of 0.01, number of epochs 10, batch size 32, gradient accumulation step 1, warmup ratio of 5%, dropout rate of 10% (mitigating overfitting), and optimizer AdamW with epsilon = 1 × 10^−8^ (we conducted experiments with varying the hyperparameters and find the optimal across all models). We found the best model over the course of the epochs.

### 2.7. Performance Evaluation

The dataset was split using the Stratified K-Fold cross-validation approach (k = 5). Model training and evaluation were performed on a NVIDIA A100-SXM4-80GB GPU (driver version 550.163.01, CUDA version 12.4).

### 2.8. Statistical Analysis

In this study, we used Macro F1-score, Sensitivity, Specificity, and AUC (Area Under the ROC Curve) as evaluation metrics, as our dataset is relatively small and exhibits class imbalance. We have chosen these metrics for the below reasons:Macro F1-score: Provides a balanced summary of model performance across all classes by equally averaging the F1-scores of each class. This approach mitigates the effects of class imbalance and ensures that minority classes are effectively captured.Sensitivity/Specificity: Established clinical metrics that quantify the model’s ability to correctly identify positive and negative cases, respectively. These metrics were chosen for their interpretability and direct clinical relevance.AUC: Measures the model’s overall discriminative ability independent of a specific decision threshold, making it particularly suitable for imbalanced or small datasets where threshold-dependent measures (e.g., PPV/NPV) can vary substantially.

A t-test was used to compare continuous variables, and one-way ANOVA was used to compare categorical variables in clinical datasets. Further statistical analysis of the LLM classification is provided in the Appendix A.

## 3. Results

### 3.1. CognoStroke Recruitment

Participant recruitment was between December 2020 and December 2024. 151 participants were recruited. Of the 151 participants recruited and consented for CognoStroke, 75 participants completed the CognoStroke assessment and 76 did not. Participants who did not participate were asked to describe reasons for non-completion. These included;

Technical challenges: Problems with web browser access and type of computer used, recording or transcript failures, participants not giving data recording permission, or not allowing use of microphones or video recording.Participant computer literacy: Not completing all elements of the assessment, feeling too fatigued, being helped by a third party, misunderstanding questions and instances where question recording ended before being fully read out.Clinical Practicalities: Recordings being interrupted by health care professionals on HASU.

### 3.2. Screening Log Review and Recruitment

As the project developed, we wanted to sample recruitment rates to identify further barriers to study recruitment. During the data collection period between November 2022 and October 2023, we monitored screening rates on HASU. A total of 1229 SSs were screened; 975 of these SSs were not suitable due to several factors including a non-stroke diagnosis, lacking capacity, no IT access, or having only limited English and significant dysarthria. Of the 254 SSs screened that were suitable for the study, 28 people were recruited (11% recruitment rate), 42 SSs did not want to take part in the study, 104 did not respond to invite letters, and 10 had visual impairment preventing completion of CognoStroke. Finally, 68 SS were eligible, but due to the busyness of HASU, they could not be contacted.

### 3.3. No Differences Between Those That Participated and Those That Left CognoStroke

We compared those that left the study after initial consent with those that remained in the study based on stroke severity, cognitive function, mood, and cardiovascular risk factors. SSs that left the CognoStroke study before assessment were not significantly different to those who completed the assessment (Table 1). SSs that did not complete CognoStroke had a mean higher number of cardiovascular risk factors. 38% had 3 or more cardiovascular risk factors compared to 24% in the participants that completed CognoStroke, although this was not statistically significant. Risk factors that were more common in non-completers included hypertension, smoking, and hypercholesterolemia.

### 3.4. CognoStroke Can Classify Cognitive Performance in SSs

Of the three LLMs evaluated using all prompts contained in the CognoStroke system (Table 2), the facebook.BART-base (BART) model gave the highest MFS score of 0.723 at a MoCA threshold of 26, GPT2 at the MoCA threshold of 25 (MFS 0.7), RobERTa-base (RobERTa) at the 24 threshold (0.645), and GPT2 at the 23 and 22 thresholds (0.664 and 0.603, respectively). We investigated to what degree individual prompts used in the CognoStroke assessment could predict cognitive impairment on their own. Figure 1 shows the distribution of MFSs for the 3 LLMs for the MoCA (cut-off 24). Interestingly, several prompts, when analysed alone, were better at classifying MoCA scores at the 24, 23, and 22 thresholds but not at 26 and 25 when compared to using all prompts together (see Figure 1 and Table 2).

We also investigated if combinations of prompts taken from the CognoStroke system could classify the different MoCA thresholds better. At each MoCA threshold, a combination of prompts improved the MFSs when compared to the whole prompt set or single prompts (see Figure 2). At MoCA thresholds of 24 and 25, GPT2 gave an MFS score of 0.805 and 0.766, respectively. Using BART at MoCA thresholds of 26, 23, and 22, MFSs of 0.783, 0.817, and 0.747, respectively, were achieved (Table 2).

### 3.5. CognoStroke Can Classify Mood Performance at the Time of Stroke

Using the same LLMs, we investigated whether CognoStroke could classify mood and anxiety based on the PHQ-9 and GAD-7 scores. For PHQ-9 the BART model predicted scores at the > 5 and > 10 thresholds with the greatest MFS (0.686 and 0.647, respectively) when using all CognoStroke prompts. When classifying GAD-7 scores using the full CognoStroke prompt set, the RoBERTa-based system delivered an MFS of 0.447 at the > 10 threshold and the BART-based system generated an MFS of 0.617 at > 5 threshold (Table 2).

Again, single prompts and combinations of prompts taken from the whole CognoStroke bank classified both GAD-7 and PHQ-9 scores better than the whole set of prompts. Using the BART system, the best single prompt gave an MFS of 0.712 for PHQ-9 of >10. The RoBERTa system gave an MFS of 0.672 for thresholds > 5. RoBERTa gave an MFS of 0.712 for GAD-7 of >10, and BART gave MFS score of 0.649 for thresholds > 5.

When using combinations of prompts, BART gave an MFS of 0.812 for PHQ-9 threshold > 10 and 0.75 for PHQ-9 threshold > 5. For GAD-7 at a threshold > 10, RoBERTa gave an MFS of 0.712, and BART gave an MFS of 0.76 at threshold > 5.

### 3.6. The BART LLM Performed Consistently Outperformed the Other LLMs

We conducted statistical significance testing on the 5-fold cross-validation results from the three models for both mood and cognition classifiers. After verifying the data distribution (using a t-test for normal distributions and a Wilcoxon test otherwise), we found that although BART consistently outperformed the others, these performance gains were only statistically significant in a few specific cases. In Table 2 we have highlighted when a specific LLM statistically out-performed the other LLMs at a particular MoCA, GAD-7, or PHQ-9 classifier (See Table 2).

Appendix A display additional MFSs for LLMs that performed inferiorly to those described in the main text.

## 4. Discussion

In this paper we have shown that automated cognitive assessment can classify both mood and cognitive scores in SSs in the HASU environment with a good level of accuracy. We have also shown that recruitment in the HASU is difficult. Factors that affect the practicalities of collecting data using an automated cognitive assessment include patient fatigue, interruptions from health care professionals, and the computer literacy of SSs.

There is limited evidence with regard to the best timing for assessing cognition and mood post stroke [18,19]. The HASU environment, due to its unselected nature, means that multiple patients do not have a stroke. HASU also caters to SSs who have significantly disabling strokes, in which cognitive assessment is less useful as deficits are often more apparent. These factors suggest that this may not be the ideal environment to perform cognitive assessment, but assessing and identifying cognitive and mood impairments at this stage allows early intervention. Although challenges exist for automated cognitive assessment in this environment, we have shown that it is feasible and that reasonable levels of accuracy can be achieved. Several of the barriers we have identified could be overcome easily by implementing regular automated cognitive assessment or reducing the length of the automated assessment to manage patient fatigue.

Previous studies have described automated cognitive assessment, but this is one of the first studies to automate mood assessment post stroke. The Cambridge Neuropsychological Test Automated Battery (CANTAB) utilises repeat serial assessment over short test–retest intervals, which have been shown to be feasible in SSs, but require a trained administrator [20]. CognoStroke offers an assessment of cognitive and mood performance that overcomes both these issues and has outputs which are familiar to stroke clinicians.

The ability to classify mood scores using CognoStroke is encouraging and offers an avenue to identify a post-stroke complication that is treatable but often missed. Mood impairment affects quality of life, and in those patients that need rehabilitation affects engagement [21]. As with cognitive impairment, mood changes post-stroke fluctuate [10]; therefore, an automated system that can track mood and identify when intervention may be needed has benefits.

Identifying that certain prompts within CognoStroke can predict cognitive and mood scores better than the full assessment is an important finding. It suggests that several of the prompts we have used in this version of CognoStroke may reduce the accuracy and future versions of the software may be improved by removing these prompts. This finding also suggests that we can develop shorter versions of CognoStroke which would benefit those with post-stroke fatigue and attention deficits. A shorter version of CognoStroke also lends itself to be used more readily in busy clinical settings, such as the TIA clinic. As described above, a large proportion of patients that present to the acute stroke pathway are often finally classified as not having a stroke. Another potential future application of a shorter version of CognoStroke would be to differentiate between acute stroke and stroke mimics [22], possibly by identifying a specific post-stroke cognitive profile. This would help triage people that present to the stroke pathway in hospital quicker, and depending on the needed length of time to perform the CognoStroke assessment, aiding in making decisions about acute stroke management such as thrombolysis and thrombectomy [23].

Our study is limited by the high screen failure rate, and even after consenting, up to 50% of participants did not complete the assessment. There is potential for selection bias in this study with the high screen failure rate, but comparable screen failure is seen in clinic trials performed in other acute stroke settings [24]. We should also consider that the digital literacy of participants may have made them more likely to partake in the study, which may also introduce bias. This study opened to recruitment early after the COVID-19 pandemic when ward visits from family and friends were restricted. We found that patients often wish to discuss study participation with family members when they are initially approached to perform a study. This negatively impacted recruitment.

An element of the failure rate after consenting could also be attributed to technical challenges of performing the automated assessment. The CognoStroke system used in this study had a relatively simple interface which required some level of computer literacy of both staff and participants to enable recording and data transfer. A better system has recently been developed that includes a tutorial and onboarding session for the participant to give a better user experience.

Future work needs to focus on identifying the optimum setting for automated cognitive assessment and whether shorter versions increase recruitment without reducing accuracy. Investigating if CognoStroke can predict cognitive and mood performance many months after stroke could help to develop appropriate treatment pathways for SSs cognitive support. Limited resources are available for cognitive rehabilitation, and an automated objective assessment could help determine who would benefit most from this therapy and could be used as an outcome measure.

## 5. Conclusions

In summary, we have found that automated assessment of mood and cognitive function is possible in the HASU setting. In this study we did see a high screen failure rate, but this is comparable to other studies performed in acute settings. Several factors exist that make assessment difficult, including digital literacy of SSs and time length of assessment, but simple changes to computer interfaces and increasing patient familiarity with this type of assessment are likely to overcome these. This study highlights how using automated assessment of mood and cognition could benefit resource-limited and highly time-pressured stroke assessment environments. Future versions of CognoStroke that focus on determining longitudinal performance on cognitive testing and uses in differentiating stroke from stroke mimics could be explored in future projects.

## Figures and Tables

**Figure 1 healthcare-13-02885-f001:**
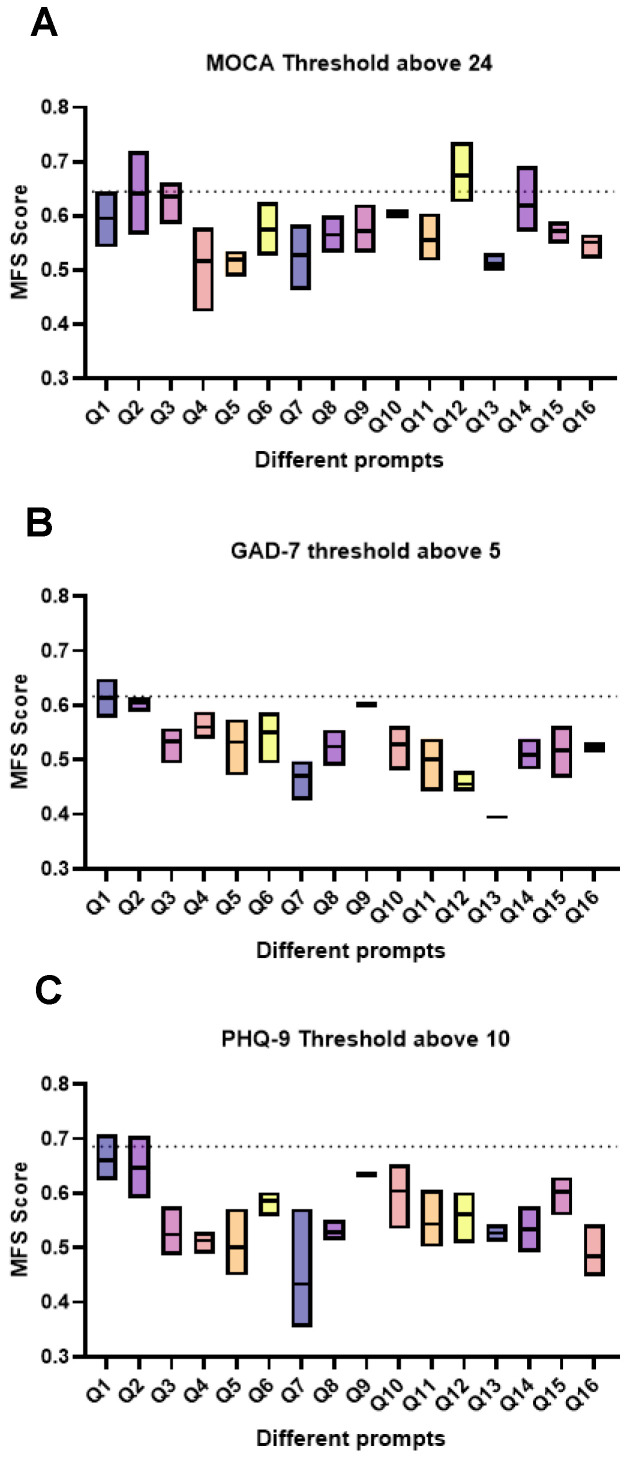
Determining the ability of separate prompts to classify different MoCA, GAD-7, and PHQ-9 thresholds: In this figure we compare how each of the separate 16 prompts that form the CognoStroke system perform in their ability to classify MoCA (**A**), GAD-7 (**B**), and PHQ-9 (**C**) thresholds. For each prompt data is displayed as the mean MFS and the range of MFS between the 3 LLMs used. The dotted line represents the MFS for the best performing LLM at this threshold when all prompts are used.

**Figure 2 healthcare-13-02885-f002:**
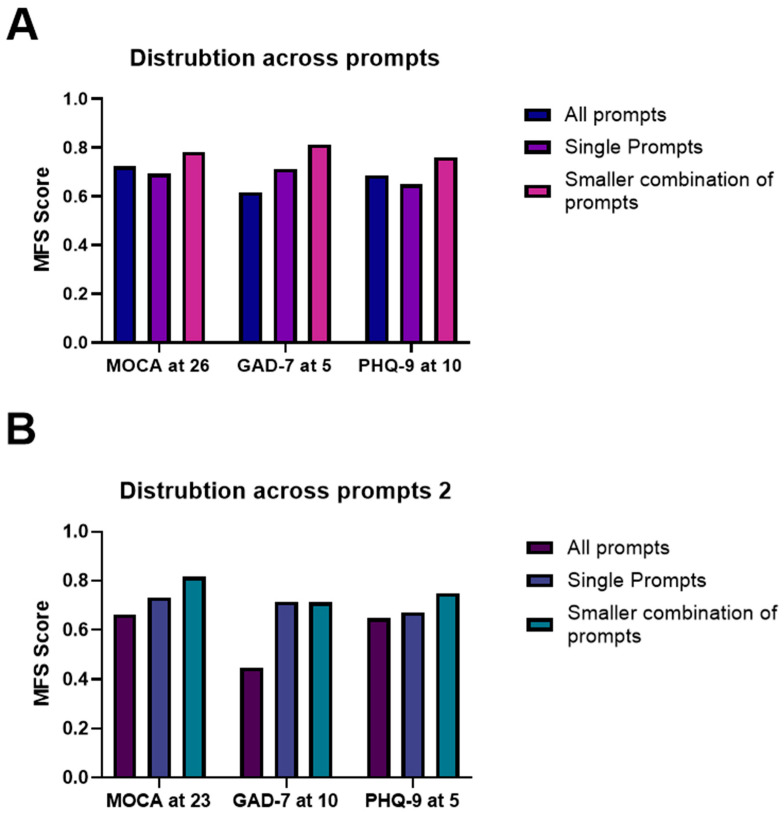
Comparison of MFSs between all, single, and smaller prompt combinations. (**A**,**B**) This figure displays for six thresholds across the two panels that a smaller combination of prompts taken from the whole CognoStroke question bank improves classification when compared to the whole set of prompts and the best performing singular prompts. The bars represent the data from the best performing LLM for each of the given thresholds and suggested number of prompts.

**Table 1 healthcare-13-02885-t001:** Demographic information for CognoStroke participants This table highlights the differences between those that stayed in the CognoStroke trial and those that left after initial consent. ~ This data has 18 missing data points. # This data has 28 data points missing. * indicates group ANOVA statistics. For GAD-7 and PHQ-9 in the group that left CognoStroke, not enough data was available to compare to those who participated in the study. National Institutes of Health Stroke Scale (NIHSS), MoCA scores, GAD7, and PHQ9 are presented as mean scores for the cohort. For participants who remained in CongoStroke, GAD-7, PHQ-9, and MoCA scores are presented with standard de-viation and range shown in brackets for each parameter.

	**Remained in CognoStroke**	**Left CognoStroke**	** *p* ** **-Value**
Total Participants	75	76	
Sex	M = 39 F = 35	M = 47 F = 29	
Age	61.3	62.1	0.488
NIHSS	3.6	3.9	0.61
Stroke Type			0.97 *
TIA	12	14	
Ischemic	52	54	
Haemorrhage	9	7	
Unclassified	1	1	
Stroke Location			0.92 *
TACS	5	6	
PACS	34	27	
LACS	12	13	
POCS	11	15	
TIA	12	14	
Risk Factors			0.263 *
Cardiovascular (MI, CCF, PVD)	5	5	
Previous stroke	13	12	
AF	9	4	
High Cholesterol	20	30	
Hypertension	29	39	
Smoker	16	25	
Obesity	5	3	
3 or more than risk factors	18	29	
Prior diagnosis Cognitive impairment	3	3	
Age Left full time education (years)	17	16.6 #	0.35
MoCA	24.7 (3.30, 15–30)	24.4 ~	0.53
Executive/Visuospatial	4	3.7 ~	0.20
Naming	2.9	2.8 ~	0.25
Attention	4.3	4.5 ~	0.58
Language	2	2.2 ~	0.22
Abstraction	1.7	1.7~	0.70
Delayed Recall	2.6	2.7 ~	0.80
Orientation	5.7	5.5~	0.15
Ethnicity	94% White British n = 69	93.5% White British n = 71	
English first language	71	72	
GAD-7	4.5 (4.56, 0–16)	NA	
PHQ-9	6.3 (5.56, 0–26)	NA	

Abbreviations: TACS, Total Anterior Circulation Stroke; PACS, Partial Anterior Circulation Stroke; LACS, Lacunar Stroke; POCS, Posterior Circulation Stroke; MI, Myocardial infarction; CCF, Congestive Cardiac Failure; PVD, Peripheral Vascular Disease; AF, Atrial Fibrillation.

**Table 2 healthcare-13-02885-t002:** Results of different LLM classification of cognitive and mood scores: Binary classification results with different thresholds on features extracted from all prompts, in terms of Macro-F1-score (MFS) (the main metric), Specificity (SP), Sensitivity (SN), and area under the curve (AUC) using the three text-based foundation models, GPT2, Facebook.BART-base (BART), and RoBERTA-base (RoBERTA). Where there is a *p* value in brackets next to a LLM this indicates that at this particular classifying threshold this particular LLM was statistically significantly better at performing the classification than the other two LLMs * = *p* < 0.05, ** = *p* < 0.01.

** *All Prompts* **						
**Outcome Measure**	**Threshold**	**Model**	**MFS**	**SP**	**SN**	**AUC**
**MoCA**	26	BART * (*p* = 0.035)	0.723	0.579	0.903	0.738
	25	GPT2	0.7	0.6	0.795	0.7
	24	RoBERTa	0.645	0.579	0.74	0.704
	23	GPT2	0.664	0.308	0.964	0.706
	22	GPT2 * (*p* = 0.0194)	0.603	0.182	0.983	0.49
**PHQ-9**	10	BART	0.686	0.86	0.5	0.676
	5	BART ** (*p* = 0.0023)	0.647	0.647	0.647	0.693
**GAD-7**	10	RoBERTa	0.447	1	0	0.476
	5	BART	0.617	0.55	0.714	0.706
** *Single Prompts* **						
**Outcome Measure**	**Threshold**	**Model**	**MFS**	**SP**	**SN**	**AUC**
**MoCA**	26	RoBERTa	0.693	0.711	0.677	0.745
	25	GPT2	0.693	0.7	0.692	0.735
	24	RoBERTa * (*p* = 0.097)	0.737	0.579	0.88	0.726
	23	BART	0.731	0.539	0.911	0.789
	22	RoBERTa * (*p* = 0.0182)	0.659	0.636	0.793	0.677
**PHQ-9**	10	BART	0.712	0.86	0.556	0.727
	5	RoBERTa * (*p* = 0.023)	0.672	0.794	0.559	0.657
**GAD-7**	10	RoBERTa	0.715	0.615	0.855	0.693
	5	BART	0.649	0.725	0.571	0.705
** *Combinations of Prompts* **						
**Outcome Measure**	**Threshold**	**Model**	**MFS**	**SP**	**SN**	**AUC**
**MoCA**	26	BART	0.783	0.711	0.871	0.79
	25	GPT2	0.766	0.769	0.767	0.791
	24	GPT2	0.805	0.632	0.94	0.78
	23	BART * (*p* = 0.028)	0.817	0.615	0.964	0.826
	22	BART	0.747	0.546	0.931	0.688
**PHQ-9**	10	BART	0.812	0.96	0.611	0.807
	5	BART * (*p* = 0.0013)	0.75	0.765	0.735	0.753
**GAD-7**	10	RoBERTa	0.712	0.927	0.462	0.718
	5	BART	0.76	0.775	0.75	0.776

## Data Availability

The data presented in this study are available on request from the corresponding author. The data are not publicly available in order to comply with the terms and conditions of the research grant awarded to the University of Sheffield.

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
