# Peer review of "CognoStroke: Automated Cognitive and Mood Assessment on the Hyper-Acute Stroke Unit"

_healthcare, 2025, doi:10.3390/healthcare13222885_

Round 1

Reviewer 1 Report

Comments and Suggestions for Authors
  • Even after participants consented to the study, up to 50% did not complete the full assessment.
  • A significant percentage of the failure rate was attributed to technical challenges. These included problems with web browser access, participants' computer literacy fatigue, and misunderstood questions.
  • Due to the presence of patients with significantly disabling strokes, The Hyper-acute Stroke Unit (HASU) may not be the ideal setting for cognitive assessment.
  • The paper does not delve into potential biases that might arise from the characteristics of the participants who completed the assessment versus the broader stroke survivor population.
  • The paper could have given a cost-effectiveness analysis to support the utility of automated assessment in resource-limited services.

Author Response

Please see the attachment." in the box if you only upload an attachment.

Reviewer 2 Report

Comments and Suggestions for Authors

I appreciate the authors’ effort in addressing an important and timely topic that integrates artificial intelligence with post-stroke assessment. However, the manuscript lacks sufficient methodological clarity, statistical rigor, and structural organization to support its conclusions. Key details on study design, data collection, model development, and ethical governance are missing or unclear, making the findings difficult to verify or interpret. I therefore recommend a major revision at this stage and encourage the authors to strengthen the study’s design, analysis, and transparency before submitting the next version.

Please view the attached file for details.

Author Response

(The authors gave the same response as above.)

Reviewer 3 Report

Comments and Suggestions for Authors

Overall

This research addresses a highly advanced and important topic: classifying patients' cognitive function (MoCA) and mood (GAD-7, PHQ-9) scores from conversational audio in the challenging environment of the acute stroke phase, using the web-based conversational agent "CognoStroke" and large-scale language models (LLMs). This approach, which seeks an automated solution to the practical issue of low assessment rates in busy clinical settings, is highly commendable.

(1) Positioning of the paper and clarification of its contribution to the field of AI

While this paper is focused on health science research, the significance of its submission to this journal and its novelty in the field are not sufficiently emphasized in the abstract and introduction.

The abstract merely highlights the tool's capabilities: "CognoStroke has potential to classify cognitive and mood scores of SSs, highlighting benefits for improving post-stroke cognitive assessment." Additionally, the introduction states that this is an update on previous research[14], stating, "In this paper, we have expanded this initial preliminary data set... We have performed a completely new analysis using three large language models (LLM)... This has led to a more complex speech pattern analysis..." However, it does not clearly state the qualitative leap or innovative contribution it represents. This makes it sound like a mere report of technical improvements. It lacks a bridging argument as to why this new approach is important to this journal, which studies online human-computer interaction, behavior, and social networks.

We suggest that the novelty of this research be more clearly asserted in the abstract and the conclusion of the introduction. "The novelty of this study lies in its demonstration, for the first time, of the feasibility of an end-to-end approach to classifying cognitive and mood scores from conversational speech by directly fine-tuning a state-of-the-art large-scale language model (LLM) for the challenging subject of acute stroke patients. This breaks away from the conventional multi-stage pipeline (ASR → feature extraction → classifier) ​​[14] and explores the possibility of more integrated and powerful automated neuropsychological assessment, providing a new methodological foundation for objective and scalable remote mental health assessment."

Furthermore, by comparing LLMs with different architectures—decoder-only (GPT2), encoder-decoder (BART), and encoder-only (RoBERTa)—this study provides valuable empirical insights into which architectures are effective in clinical score classification tasks based on conversational speech." Shouldn't we also clarify its contribution as AI research?

(2) The Appropriateness of the AI ​​Method Selection and Comparative Review of Methodologies

Your reference to "SHAP" appears to be a confusion with other reviews, but your underlying point that the appropriateness of the adopted methodology and the reasons for its selection are unclear fully applies to this paper. In the "Automated Language Processing" section of the methodology, it states, "In the current study, we replaced the ASR component with a large Whisper model... and, instead of using BERT embeddings, we directly fine-tuned large language models (LLMs)... Three LLMs (GPT2..., facebook.BART-base..., and RobERTa-base...) were used..."

Please explain the theoretical motivation for replacing the Wav2vec2+BERT+classifier pipeline with direct fine-tuning of LLM. For example, wouldn't it be necessary to consider something like, 'Direct fine-tuning enables end-to-end training of task-specific representations, potentially capturing more nuanced linguistic features related to cognitive and mood states than pre-trained, fixed BERT embeddings.'

(3) Insufficient explanation of dataset and preprocessing

As you point out, the explanation of the dataset and preprocessing is very brief.

Quotation from the paper (data collection): "Cognitive performance, mood, and anxiety levels were assessed using GAD-7, PHQ-9, and MoCA or telephone (t-MoCA) scales." "t-MoCA scores were scaled to fit full MoCA results prior to analysis." This alone does not provide any insight into what data was collected or how it was processed. Basic information is missing, such as how the t-MoCA scores were adjusted and what preprocessing (e.g., noise removal, silent segment removal) was performed on the speech data before inputting it into Whisper. For the 75 subjects ultimately analyzed, please supplement the information in Table 1 and describe the distributions (mean, standard deviation, and range) of MoCA, GAD-7, and PHQ-9 scores. Would this help readers understand the difficulty of each classification task and the degree of class imbalance?

(4) Recognizing Issues in Presentation of Results and Conclusions

Table 2 shows the performance of three different LLMs for three different clinical scales (MoCA, PHQ-9, and GAD-7) at multiple cutoff points for three cases: "all questions," "single question," and "combination of questions," using four different metrics (MFS, SP, SN, and AUC). This is information-dense, making it extremely difficult to decipher the paper's most important findings. The conclusion (Section 5. Conclusions) is very concise. Important limitations mentioned in the Discussion (Section 4), such as the high dropout rate (consenting, up to 50% of participants did not complete the assessment), are not sufficiently highlighted in the conclusions.

The presentation of the results prevents readers from grasping the key message. Shouldn't the conclusions not just summarize the successes of the study, but also honestly identify its limitations and future challenges?

Table 2 contains too much information and obscures the paper's key findings. We strongly recommend a fundamental rethinking of the structure of the results. For each clinical scale (MoCA, PHQ-9, GAD-7), present the key findings in simpler tables and annotated graphs. For example, create bar graphs comparing the best MFS achieved with the "all questions," "single question," and "combined questions" approaches, clearly indicating which model and threshold were optimal. Doesn't the current format obscure the most important findings of this study?

Author Response

(The authors gave the same response as above.)
